# LOST: Low-rank and Sparse Pre-training for Large Language Models

## Abstract

While large language models (LLMs) have achieved remarkable performance across a wide range of tasks, their massive scale incurs prohibitive computational and memory costs for pre-training from scratch. Recent studies have investigated the use of low-rank parameterization as a means of reducing model size and training cost. In this context, sparsity is often employed as a complementary technique to recover important information lost in low-rank compression by capturing salient features in the residual space. However, existing approaches typically combine low-rank and sparse components in a simplistic or ad hoc manner, often resulting in undesirable performance degradation compared to full-rank training. In this paper, we propose **LO**w-rank and **S**parse pre-**T**raining (**LOST**) for LLMs, a novel method that ingeniously integrates low-rank and sparse structures to enable effective training of LLMs from scratch under strict efficiency constraints. LOST applies singular value decomposition to weight matrices, preserving the dominant low-rank components, while allocating the remaining singular values to construct channel-wise sparse components to complement the expressiveness of low-rank training. We evaluate LOST on LLM pretraining ranging from 60M to 7B parameters. Our experiments show that LOST achieves competitive or superior performance compared to full-rank models, while significantly reducing both memory and compute overhead. Code will be made available.

## 1 Introduction

Large language models (LLMs) have demonstrated remarkable achievements across various domains. However, due to the billions of parameters and the pretraining-finetuning paradigm, LLMs typically require substantial memory and computational resources Samsi et al. (2023), which is a longstanding obstacle for their applications. In terms of fine-tuning, low-rank approximation, pioneered by LoRA Hu et al. (2021), has gained popularity due to its high effectiveness. Instead of updating full metrics, LoRA fine-tunes only the low-rank adaptors while keeping the pre-trained weights frozen, significantly reducing the memory usage and computational costs. Following LoRA, numerous LoRA variants have been proposed to improve its efficacy and efficiency, including but not limited to (Zhang et al., 2023; Renduchintala et al., 2023; Sheng et al., 2023; Liu et al., 2024; Kopiczko et al., 2023; Dettmers et al., 2024).

While existing methods have advanced our understanding of efficient LLM fine-tuning, their effectiveness in the context of LLM **pre-training**—a substantially more resource-intensive stage—remains largely underexplored. Prior efforts to train neural networks from scratch with low-rank structures have been primarily limited to small-scale models (Khodak et al., 2021; Saada & Tanner, 2023; Kamalakara et al., 2022), dependent on full-rank warmup training (Lialin et al., 2023; Jaiswal et al., 2024), restricted to feed-forward network (FFN) layers (Wei et al., 2024; Fernandez-Lopez et al., 2025), or involve full-rank weights updated via low-rank gradients (Zhao et al., 2024; Chen et al., 2024; Zhu et al., 2024; Zhang et al., 2024). Despite their promise for improving efficiency, low-rank weight pre-training of LLMs consistently underperforms compared to full-rank training (Zhao et al., 2024; Han et al., 2024).

In this paper, we propose **LO**w-rank and **S**parse **T**raining (**LOST**) for LLMs, which enables training LLMs with low-rank weights while maintaining performance as good as full-rank training (see its performance in Figure 1).

**Technical Novelty:** ① While there exist previous works exploring the co-design of low-rank and sparsity (Li et al., 2023; Huang et al., 2025; Ding et al., 2023), Singular Value Decomposition (SVD) for low-rank initialization (Meng et al., 2025; Bałazy et al., 2024; Lin et al., 2024; Paischer et al., 2024), they primarily focus on fine-tuning, rather than pre-training, which is a more challenging scenario; ② Recently, SLTrain (Han et al., 2024) incorporates both low-rank structures and sparsity during pre-training. However, SLTrain combines low-rank and sparse components naively with independent initialization, ignoring the complex interaction between the two components. In contrast, we co-design the low-rank and sparse components to **complement each other**, aiming to

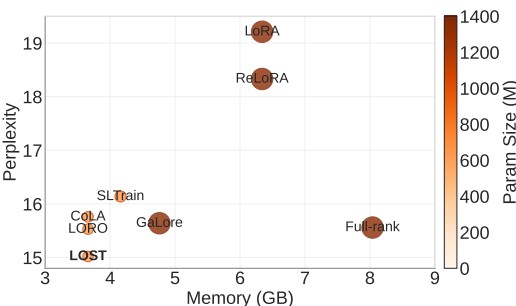

Figure 1: Performance comparison of pretraining methods on LLaMA-1B (C4 dataset). Smaller circles in the lower-left indicate better memory efficiency and lower perplexity. See Table 1 for complete results.

preserve the desirable presentation capacity and trainability of full-rank models. Specifically, we initialize the low-rank module using SVD[1] of full-rank initialization to capture the dominant subspace associated with the largest singular values, and complement it with a sparse residual matrix that preserves information in the remaining subspace orthogonal to the dominant subspace. The goal of this design is to retain the essential favorable properties of full-rank training, which is typically associated with optimal performance. Our results demonstrate that **LOST** outperforms previous methods that integrate sparsity with low-rank modeling. Figure 2 provides an overview of the LOST procedure. To summarize, our main contributions are outlined below:

① We propose LOST, a novel low-rank training approach that combines low-rank and sparse components to enable efficient LLM pre-training from scratch. This method achieves parameter and memory efficiency by eliminating the need for full-rank pre-training while maintaining optimal performance.

② Our main novelty lies in the co-design of low-rank and sparse components that complement each other: the low-rank part captures the dominant subspace associated with the largest singular values, while the sparse residual preserves information in the remaining subspace. This complementary design enables us to retain the desirable properties of full-rank training.

③ We validate LOST by pre-training LLaMA models of various sizes (from 60M to 7B) from scratch on the C4 dataset, demonstrating performance and efficiency improvements. We further show strong generalizability of LOST to fine-tuning tasks on LLMs.

## 2 BACKGROUND

### 2.1 LOW RANK AND SPARSE DECOMPOSITION

Low-rank approximation has emerged as a prominent approach to parameter-efficient LLM finetuning by decomposing the full-rank weight matrix $W$ into a product of two low-rank factors (denoted as $A$ and $B$) Hu et al. (2021). Despite their widespread adoption, low-rank approximation methods generally suffer from performance degradation compared to full-rank fine-tuning Lialin et al. (2023); Hu et al. (2021). This performance gap is typically attributed to the reduced number of trainable parameters and underlying factors, such as altered gradient dynamics and training dynamics Nasiri & Garraghan (2025); Kamalakara et al. (2022). To alleviate these issues, low-rank approximation has been integrated with complementary model compression techniques, such as quantization Dettmers et al. (2023); Xu et al. (2023) and pruning Chen et al. (2023); Li et al. (2023); Chen et al. (2022) techniques. Among them, sparse plus low-rank decomposition that approximates model weights as the sum of sparse and low-rank matrices has emerged as a promising direction in LLM compression Zhang & Papyan (2024); Candès et al. (2011). While this methodology (also known as robust PCA) has been well-studied across various domains Candès et al. (2011), its recent application to LLMs has revealed significant potential for enhancing fine-tuning efficiency. For example, Zhang & Papyan

---

[1]We perform SVD only once for our initialization before training, which introduces no overhead for training.

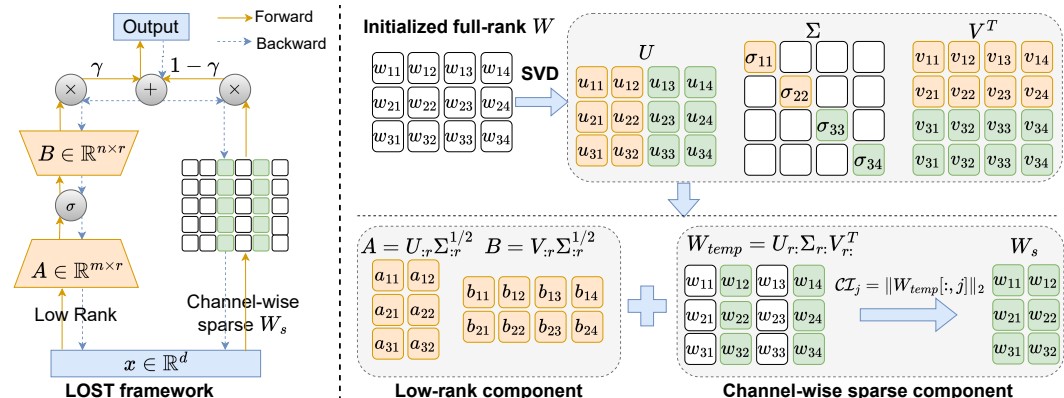

Figure 2: Framework of the LOST method. LOST begins by initializing a full-rank weight matrix $W$ and performing SVD on $W$, from which it extracts the top-$r$ singular values and vectors to form $W_l = AB^T$. A temporary weight matrix $w_{comp}$ is constructed using the remaining singular vectors and values, which guides the creation of a sparse mask $M$ and the sparse matrix $W_s$. The final reconstructed matrix combines both low-rank and sparse components, with their relative contributions controlled by a trade-off coefficient $\gamma$.

(2024) proposed OATS to decompose LLMs with SVD for low-rank structure and outlier information for sparsity. Advanced sparse matrices can be used, such as Butterfly in Chen et al. (2022). More recently, Makni et al. (2025) proposed a new optimization-based framework for better model utility-compression tradeoffs. It is worth noting that most existing low-rank and sparse decomposition approaches focus on fine-tuning scenarios, still requiring the notoriously expensive pretraining. This limitation motivates our exploration of low-rank and sparse methods for training LLMs from scratch.

## 2.2 Low-rank Pretraining

While some studies investigated training neural networks from scratch with low-rank structures, they have been limited to small-scale models or only the feed-forward networks (FFN) layers within language models Khodak et al. (2021); Kamalakara et al. (2022); Saada & Tanner (2023); Wei et al. (2024). Hence, low-rank LLM pertaining is acknowledged as crucial yet challenging. Recently, a few attempts have been done to achieve low-rank pertaining for LLMs. ReLoRA Lialin et al. (2023) allows training low-rank models from scratch by using a warm-up phase based on full-rank models. Another appealing approach is to use low-rank updates to support the full-rank training of LLMs. For example, GaLore Zhao et al. (2024) projects gradients into low-rank subspaces to achieve low-rank updates and memory efficiency, which, however, is not parameter efficient. A follow-up work, Q-GaLore Zhang et al. (2024), further reduces memory overhead using quantization. More recently, LORO Mo et al. (2025) allows the low-rank factors to be jointly updated by using Riemannian optimizer. An orthogonal work, COLA Liu et al. (2025), focuses on low-rank activations and adopts non-linear activation between factorized weight matrices. The most closely related to our work is SLTrain Han et al. (2024), which combines low-rank and unstructured sparse components to enhance low-rank pretraining for LLMs. SLTrain adopts a commonly used LoRA type of initialization, i.e., Kaiming initialization He et al. (2015), and uniform initialization for the low-rank and unstructured sparse components, respectively, without exploring the use of complementary information between them. Unlike SLTrain, our approach LOST generates the low-rank and structured sparse components in a complementary manner, thereby enhancing the model's expressiveness.

## 3 Methodology

In this section, we introduce Low-rank and Sparse Training (LOST), our proposed method for efficient model training. As shown in Figure 2, LOST begins by initializing a full-rank weight matrix $W \in \mathbb{R}^{m \times n}$ using standard initialization techniques He et al. (2015). This matrix is then decomposed into low-rank and sparse components. Unlike traditional low-rank plus sparse approaches that combine them directly Han et al. (2024), we leverage Singular Value Decomposition (SVD) to ensure

that the low-rank and sparse components complement each other in orthogonal rank subspaces. To enable hardware acceleration, the sparse component is structured in a channel-wise manner. Furthermore, a non-linear activation function is inserted between the low-rank matrices to enhance the model's expressiveness. Below, we detail the procedure for constructing the low-rank and sparse components.

## 3.1 LOW-RANK MODELING

To effectively find the principal low-rank component $W_l$, we perform SVD on the initialized full-rank $W \in \mathbb{R}^{n \times n}$:

$$SVD(W) = U\Sigma V^T = \sum_{i=1}^{\text{rank}(W)} \sigma_i u_i v_i^T,  \tag{1}$$

where $U = [u_1, \ldots, u_m] \in \mathbb{R}^{m \times m}$ and $V = [v_1, \ldots, v_n] \in \mathbb{R}^{n \times n}$ are the left- and right-singular vector matrix, respectively, and $\text{diag}(\sigma_1, \ldots, \sigma_n) \in \mathbb{R}^{m \times n}$ is the singular value matrix with $\sigma_1 \geq \sigma_2 \geq \ldots \geq 0$. Given the target rank $r$, we select the top-$r$ singular values and their corresponding singular vectors to construct a low-rank approximation $W_l = AB^T$ as:

$$A = U_r \Sigma_r^{1/2} = \left[\sigma_1^{1/2} u_1, \ldots, \sigma_r^{1/2} u_r\right] \in \mathbb{R}^{m \times r},$$

$$B = V_r \Sigma_r^{1/2} = \left[\sigma_1^{1/2} v_1, \ldots, \sigma_r^{1/2} v_r\right] \in \mathbb{R}^{n \times r}.$$

where $\Sigma_r$, $U_r$, and $V_r$ represent the retaining top-$r$ singular values and the corresponding singular vectors, respectively. Following Liu et al. (2025), we add SiLU non-linear activation between $A$ and $B$ to enhance performance.

This SVD-based approach provides optimal low-rank approximation under the Frobenius norm while naturally yielding a factorized form for $W_l$, reducing the number of parameters from $mn$ to $r(m + n)$. However, the truncation of smaller singular values in $W_l$ inevitably results in information loss, with greater losses occurring at smaller $r$. This leads to degraded performance since SVD does not consider the relative importance of weights Wang et al. (2024).

## 3.2 CHANNEL-WISE SPARSE MODELING

Although SVD preserves the principal subspace of the weight matrix, relying solely on low-rank modeling may lead to limited expressivity Hsu et al. (2022). Ideally, we aim to retain both the dominant subspaces associated with large singular values and the bulk subspaces corresponding to smaller ones, as together they are essential for maintaining the full representational capacity and optimal trainability of LLMs. To do so, we introduce our channel-wise sparse metrics derived from the remaining subspaces using smaller singular values in Eq. 1. Concretely, we select channels from the remaining subspace represented by $W_{comp} = U_{r:} \Sigma_{r:} V_{r:}^T = \sum_{i=r+1}^{\text{rank}(W)} \sigma_i u_i v_i^T$. Given a target sparsity ratio $\rho$ for $n$ overall channels, we identify $k = \lceil \rho \cdot n \rceil$ channels to retain from $W_{comp}$ via $L_2$-norm channel-wise importance score $\mathcal{CI}$:

$$\mathcal{CI}_j = \|W_{comp}[:,j]\|_2, \quad j = 1, \ldots, n.  \tag{2}$$

The top-$k$ channels with the highest importance scores are selected, with their indices denoted as $\mathcal{I} = \text{argsort}(\mathcal{CI})[-k:]$. Accordingly, the sparse weight matrix is constructed as:

$$W_s = W[:, \mathcal{I}] \in \mathbb{R}^{m \times k},  \tag{3}$$

which only stores the weights corresponding to the selected channels.

Previous element-wise sparsity used in SLTrain Han et al. (2024) requires storing a binary mask or the int64 indices, resulting in twice the memory of the sparse component itself. Channel-wise sparsity retains entire input channels (columns) and significantly reduces storage requirements. While this approach still requires storing the indices of the selected channels, the associated memory cost is negligible compared to that of element-wise sparsity, as the number of channels is much smaller than the number of individual elements.

---

**Algorithm 1:** Low-rank and Sparse Training for LLMs (LOST)

---

**Input:** $W$: initial weight matrix, $r$: target rank for low-rank approximation, $\rho$: target sparsity
ratio, $\gamma$: trade-off coefficient, $\sigma$: activation function;

**Step 1: Low-rank components initialization**

Perform SVD on $W$: $W = U\Sigma V^T$;

$W_l = AB^T$, $A = U_r\Sigma_r^{1/2}$, $B = V_r\Sigma_r^{1/2}$;

**Step 2: Sparse components initialization**

$W_{comp} = U_{r:}\Sigma_{r:}V_{r:}^T$;          $\triangleright$ Compute complementary matrix with remaining singular values.

$\text{importance}_j = \|W_{comp}[:,j]\|_2, \quad j = 1, \ldots, n$;          $\triangleright$ Calculate channel importance score.

$\mathcal{I} = \text{argsort}(\text{importance})[-k:]$;          $\triangleright$ Select top-$k$ channels.

$W_s = W[:, \mathcal{I}] \in \mathbb{R}^{m \times k}$;

**Step 3: Training**

**for** *each training iteration* **do**

  Forward: $o = \gamma \cdot \sigma(xA)B^T + (1 - \gamma) \cdot x_{[:,\mathcal{I}]}W_s^T$;          $\triangleright$ Combine the outputs together.

  Backward: Update $A$, $B$ and $W_s$ through gradient descent;

**Output:** Weight matrix $\hat{W}$

---

### 3.3 TRAINING PROCESS

During forward propagation, the computation is performed efficiently by combining the activated low-rank and channel-wise sparse components:

$$o = \gamma \cdot \sigma(xA)B^T + (1 - \gamma) \cdot x_{[:,\mathcal{I}]}W_s^T, \tag{4}$$

where $x$ represents the input, $x_{[:,\mathcal{I}]}$ selects only the channels specified by $\mathcal{I}$. The coefficient hyperparameter $\gamma \in [0, 1]$ controls the relative importance of the two components. This method can be adapted to different tasks by adjusting the rank of the low-rank components, the sparsity level of the sparse components, and the trade-off coefficient. Algorithm 1 outlines the pseudocode of LOST.

For back propagation, the weights of the low-rank components and the sparse components are updated with gradient descent. This decomposition method is implemented consistently across all linear layers within the attention mechanism and feed-forward MLP layers throughout the transformer architecture, ensuring a uniform approach to parameter efficiency.

### 3.4 MEMORY AND COMPUTATIONAL ANALYSIS

LOST achieves significant parameter reduction through the combination of activated low-rank approximation and channel-wise structured sparsity. While the low-rank component requires $r(m+n)$ parameters, the sparse component requires $mk$ parameters for the selected columns plus $k$ indices, where $k = \lceil \rho \cdot n \rceil$ and $k \ll \min(m, n)$. Overall, LOST reduces the total parameters from $mn$ to $r(m + n) + mk$. The memory efficiency of LOST is enhanced by the use of channel-wise sparsity, which eliminates the substantial overhead associated with storing binary masks or element-wise indices. Moreover, such a structured sparsity approach maintains efficient memory access patterns during inference. Noting that, while using an activation function between the two low-rank matrices $A$ and $B$ adds negligible computational overhead, it enhances LOST's model expressiveness and representational ability.

## 4 EXPERIMENTS

We conduct experiments on LLM pre-training and fine-tuning, and a series of ablation studies to validate the effectiveness of LOST. All experiments were conducted on NVIDIA A100/H100 GPUs.

### 4.1 LLMs PRE-TRAINING

**Experimental setup.** We conduct LLM pre-training on the Colossal Clean Crawled Corpus (C4) dataset Raffel et al. (2020). The C4 dataset is a large-scale collection of web-crawled texts that has

Table 1: Comparison of perplexity, parameter count in millions (Param), and estimated memory consumption in gigabytes (G) across different methods. $r$ and $d$ denote the target rank and the hidden dimension of the LLM model, respectively. LOST uses an actual rank lower than r to ensure the parameter count does not exceed other baseline methods. Results of LORO and CoLA are reproduced by using the their default scripts. Results for other methods are directly reported from Zhao et al. (2024); Han et al. (2024).

| | 60M | | | 130M | | | 350M | | | 1B | | |
|---|---|---|---|---|---|---|---|---|---|---|---|---|
| $r/d$ | 128/512 | | | 256/768 | | | 256/1024 | | | 512/2048 | | |
| Tokens | 1.1B | | | 2.2B | | | 6.4B | | | 13.1B | | |
| Method | PPL↓ | Param(M) | Mem(G) | PPL↓ | Param(M) | Mem(G) | PPL↓ | Param(M) | Mem(G) | PPL↓ | Param(M) | Mem(G) |
| Full-Rank | 34.06 | 58 | 0.35 | 24.36 | 134 | 0.81 | 18.80 | 368 | 2.21 | 15.56 | 1339 | 8.04 |
| LoRA | 34.99 | 58 | 0.36 | 33.92 | 134 | 0.84 | 25.58 | 368 | 1.85 | 19.21 | 1339 | 6.34 |
| ReLoRA | 37.04 | 58 | 0.36 | 29.37 | 134 | 0.84 | 29.08 | 368 | 1.85 | 18.33 | 1339 | 6.34 |
| GaLore | 34.88 | 58 | 0.28 | 25.36 | 134 | 0.61 | 18.95 | 368 | 1.59 | 15.64 | 1339 | 4.76 |
| LORO | 33.87 | 43 | 0.24 | 24.78 | 94 | 0.57 | 19.66 | 185 | 1.11 | 15.53 | 609 | 3.66 |
| CoLA | 34.10 | 43 | 0.24 | 25.61 | 94 | 0.57 | 19.75 | 185 | 1.11 | 15.76 | 609 | 3.66 |
| SLTrain | 34.15 | 44 | 0.26 | 26.04 | 97 | 0.60 | 19.42 | 194 | 1.24 | 16.14 | 646 | 4.16 |
| **LOST** | **32.25** | 43 | 0.24 | **24.05** | 94 | 0.57 | **18.95** | 185 | 1.11 | **15.02** | 609 | 3.66 |

been extensively cleaned and filtered, and is widely adopted for model pre-training. Following the experimental settings in Han et al. (2024), we train each model for a single epoch over the training split of the dataset.

We use Llama-based architectures with model sizes ranging from 60M to 7B parameters Touvron et al. (2023). Our implementation includes pre-normalization, RMSnorm, and SwiGLU activation functions Zhang & Sennrich (2019); Shazeer (2020). We adhere closely to established protocols outlined in recent literature Zhao et al. (2024); Han et al. (2024), including using the BF16 format to enhance memory efficiency. We also adopt the optimizer configurations, cosine learning rate decay, and warmup strategies as detailed in Zhao et al. (2024); Han et al. (2024). The detailed parameter configurations for models of different sizes are presented in Table 12.

**Baseline.** We compare our method with standard baselines, i.e., Full-Rank that performs pretraining with a full-rank model, LoRA Hu et al. (2021), and state-of-the-art pre-training approaches, including ReLoRA Lialin et al. (2023), GaLore Zhao et al. (2024), LORO Mo et al. (2025), CoLA Liu et al. (2025), and SLTrain Han et al. (2024). We ensure a fair comparison based on the same training token number.

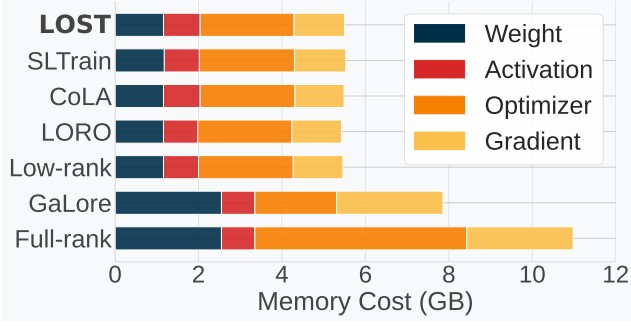

Figure 3: Breakdown of memory consumption across different methods on 1B model.

**Hyperparameters.** For all sizes of the Llama-based models trained with LOST, we set the coefficient $\gamma$ to 0.7, and use a rank of 256 when applying SVD-based initialization for the sparse components. Across all models, we configure LOST with a sparsity level of 0.01 for the sparse component and correspondingly adjust the low-rank component's rank to maintain a parameter count comparable to other baselines.

**LOST demonstrates state-of-the-art results in low-rank weight pre-training, surpassing previous approaches.** Table 1 shows that LOST outperforms all baseline methods across all model

sizes in terms of perplexity without incurring additional memory overhead. Compared to full-rank models, LOST achieves comparable performance at 350M parameters while surpassing full-rank models at other scales. Specifically, LOST achieves 3.5% lower perplexity than the full-rank model at 1B, 1.2% lower at 130M, and 5.3% lower at 60M parameters. The comparison between LOST and its closest baseline, SLTrain, illustrates significant improvements in both performance and memory efficiency across all model scales (also can be observed in Figure 1), showcasing the effectiveness of the proposed low-rank and sparse training strategy.

**LOST delivers strong efficiency gains.** We compare the actual memory consumption and breakdown of different methods on the 1B model. We set the batch size to 1 to clearly display memory usage across different components, including weight, activation, gradient, and optimizer states. For the compared methods (Full-rank, Low-rank, CoLA, SLTrain, and LORO), we maintain their default configurations as in Zhao et al. (2024); Han et al. (2024); Mo et al. (2025); Liu et al. (2025). We use *bfloat16* data type and disable gradient checkpointing for the memory consumption estimation, and the results are illustrated in Figure 3. We can observe that all low-rank methods, including LOST, achieve significant memory reduction, decreasing memory usage by nearly half compared to the full-rank model. We anticipate that the memory efficiency benefits will be further amplified when using larger batch sizes.

**Scaling performance.** To evaluate LOST's scalability to larger model sizes, we conducted experiments with the LLaMA-7B model on 8× NVIDIA H100 GPUs. As shown in Table 2, we conduct LOST with both standard and 8-bit optimizers. Due to computational constraints, we trained the model for only 40K steps instead of the full training schedule. LOST outperforms 8-bit Adam and 8-bit GaLore at the same number of steps. The 8-bit version of LOST further reduced memory usage while maintaining competitive performance. According to Liu et al. (2025), removing certain activation functions from the base model might further improve performance. In this work we didn't study this as we focused on evaluating LOST with the standard model architecture.

## 4.2 ABLATION STUDY

Table 2: Validation perplexity and actual memory footprint per GPU were reported for the LLaMA-7B model pre-trained on the C4 dataset for 40K steps. Baseline results are collected from Zhao et al. (2024); Han et al. (2024). LOST uses Adam optimizer and 8-bit LOST uses 8-bit Adam optimizer.

| Method | Mem (G) | 10K | 40K |
|---|---|---|---|
| 8-bit Adam | 72.59 | N/A | 18.09 |
| 8-bit GaLore | 65.16 | 26.87 | 17.94 |
| 8-bit SLTrain | 60.91 | 27.59 | N/A |
| LOST | 62.15 | **24.41** | **16.48** |
| 8-bit LOST | 50.19 | 24.67 | 17.59 |

A series of ablation studies is conducted on the LLaMA-60M and LLaMA-130M models due to the limited computing resources.

**Ablation on the complementarity between $W_s$ and $W_l$.** To evaluate the effectiveness of our proposed channel-wise sparse component $W_s$ in compensating for the truncation loss from low-rank factorization, we investigate the impact of different channel selection methods on 60M and 130M models with a sparsity of 0.01. Our comparison examines two sources for the complementary matrix $W_{comp}$: 1) SVD-based: $rem$ (remaining singular values after truncating $W_l$, used in LOST), $top$ (largest singular values), $bot$ (smallest singular values), and $rand$ (random singular values). 2) Non-SVD based: $INI$ (initialized full-rank matrix). To select channels from the complementary matrix, we apply three selection criteria: L1-norm, L2-norm and random selection (denoted as $l1$, $l2$, $rand$ subscripts).

As shown in Table 3, our $SVD_{l_2}^{rem}$ shows constantly better performance than others for both the 60M and 130M models. For the complementary matrix, $SVD$ generally performs better than $INI$, verifying the advantage of the co-design of $W_l$ and $W_S$. Notably, $rem$ achieves better performance compared to $top$, $bot$ and $rand$ alternatives, due to its strategic selection of complementary singular values. These results demonstrate the effectiveness of LOST.

**Ablation on the initialization of low-rank matrices $A$ and $B$.** To investigate the impact of the **SVD** based initialization of $A$ and $B$, we tested LOST with different initialization strategies: **Kaiming** Han et al. (2024) initializes matrix $A$ using Kaiming initialization and matrix $B$ with zeros; **Xavier** Mo et al. (2025) initializes both $A$ and $B$ using Xavier initialization; **CoLA-style** Liu et al. (2025)

Table 3: The impact of different channel-wise sparse components generated by different strategies. **SVD** based methods first perform SVD decomposition with $k = 256$ singular values, and then generate $W_{comp}$, from which select channels. $\textbf{SVD}_{\textbf{l}_2}^{\textbf{rem}}$ is used by LOST.

| Method | $SVD_{l_2}^{rem}$ | $SVD_{l_2}^{bot}$ | $SVD_{l_2}^{top}$ | $SVD_{l_2}^{rand}$ | $SVD_{l_2}^{rand}$ | $SVD_{l_1}^{rem}$ | $SVD_{l_1}^{bot}$ | $SVD_{l_1}^{top}$ | $INI_{l_2}$ | $INI_{l_1}$ | $INI_{rand}$ |
|---|---|---|---|---|---|---|---|---|---|---|---|
| 60M | **32.25** | 32.31 | 32.41 | 32.35 | 32.39 | 32.33 | 32.29 | 32.37 | 32.34 | 32.46 | 32.40 |
| 130M | **24.01** | 24.13 | 24.28 | 24.15 | 24.27 | 24.14 | 24.17 | 24.18 | 24.24 | 24.17 | 24.34 |

initializes both matrices using Gaussian distributions with variance computed based on rank and dimension values. The PPL results are summarized in Table 4, and demonstrate the advantages of **SVD** over its peers in both model sizes. The superior performance of **SVD** can be attributed to its ability to preserve the spectral properties of the original weight matrix. Unlike **Kaiming** and **Xavier** initialize $A$ and $B$ independently without considering their interaction, **SVD** ensures that the product $AB^T$ optimally approximates the original full-rank weights in the Frobenius norm sense. Although **CoLA-style** initialization benefits from the variance scaling based on rank and dimension, it still underperforms **SVD**. These observations highlight the importance of preserving the inherent structure of the weight matrix, thus validating our design choice of using SVD-based initialization in LOST.

Table 4: The impact of initialization methods for low-rank components.

| Method | SVD | Kaiming | Xavier | CoLA-style |
|---|---|---|---|---|
| 60M | **32.25** | 33.13 | 33.03 | 32.71 |
| 130M | **24.01** | 24.93 | 24.77 | 24.70 |

Table 5: Ablation on how to combine low-rank and sparse components.

| Method | Weight Avg | Output Avg |
|---|---|---|
| 60M | 33.70 | 33.62 |
| 130M | 24.62 | 24.79 |

**How to combine of low-rank and sparse components.** We investigate two strategies for combining low-rank and sparse components: **Weight Avg** $W = \gamma W_l + (1 - \gamma)W_s$ and **Output Avg** $y = \gamma y_l + (1 - \gamma)y_s$ combine the low-rank and sparse components at the weight and output levels, respectively. As shown in Table 5, while weight-level averaging is a common choice, output-level averaging shows comparable results. The key advantage of the output-level combination lies in our use of activation functions between the low-rank matrices.

**Ablation on the activation between low-rank components.** We examine the impact of the use of activation functions between the low-rank matrices. Table 6 compares the performance with and without the activation. The results demonstrate that including activation functions improves performance across both model sizes. The activation function introduces non-linearity between the low-rank factors, enhancing the model's expressiveness without adding parameters.

**Ablation on the rank number for the complementary matrix $W_{comp}$.** The rank values for constructing $W_{comp}$ determine the number of singular values used to for channel selection. Table 7 shows results with rank values ranging from 32 to 512. We can see that our SVD-based sparse initialization is robust to this hyperparameter. Our default choice of rank 256 provides good performance for both model sizes while balancing computational efficiency.

Table 6: Impact of the activation between low-rank weight matrices.

| Method | Activated | Non-activation |
|---|---|---|
| 60M | **32.25** | 33.51 |
| 130M | **24.01** | 24.83 |

Table 7: The impact of different rank numbers when constructing $W_{comp}$.

| Rank_number | 32 | 64 | 128 | 256 | 512 |
|---|---|---|---|---|---|
| 60M | 32.15 | 32.33 | 32.29 | **32.25** | 32.51 |
| 130M | 24.28 | 24.21 | 24.17 | **24.01** | 24.17 |

**Impact of parameter allocation between $W_l$ and $W_s$ with fixed budgets.** Given a fixed parameter budget, we investigate how different allocations of parameters between low-rank and sparse components affect model performance. Table 8 shows the results with varying sparsity levels from 0.01 to 0.3, where the rank of the low-rank component is adjusted accordingly to maintain constant total parameters. The results reveal that lower sparsity levels achieve optimal performance. As sparsity increases beyond 0.1, performance degrades significantly. This suggests that allocating more parameters to the low-rank component while maintaining a highly selective sparse component yields better results. Note that these experiments use a fixed $\gamma = 0.7$, which may not be optimal for higher sparsity settings, as $\gamma$ may need to be adjusted to effectively balance the contributions of the two

components. This observation supports our default choice of low sparsity (0.01) in LOST, where the sparse component serves as a targeted complement rather than a primary contributor.

Table 8: Impact of parameter allocations on model performance with a fixed parameter budget.

| Sparsity | 0 | 0.01 | 0.05 | 0.1 | 0.2 | 0.3 |
|----------|-------|-------|-------|-------|-------|-------|
| 60M | 32.93 | 32.25 | 32.80 | 33.50 | 36.53 | 42.79 |
| 130M | 24.74 | 24.01 | 24.29 | 24.80 | 26.10 | 27.61 |

**Ablation on the coefficient $\gamma$.** The trade-off coefficient $\gamma$ controls the relative importance of low-rank and sparse components in the output combination. Table 9 shows performance across different $\gamma$ values. The results indicate that allocating 70-80% weight to the low-rank component at the target sparsity (0.01) yields optimal performance, confirming that the low-rank component captures most of the essential information while the sparse component provides complementary yet critical refinements. Notably, we keep $\gamma$ as a fixed hyperparameter rather than a learnable parameter, as making $\gamma$ trainable would require storing intermediate activations for gradient computation, leading to substantial memory overhead that contradicts our efficiency goals.

Table 9: Ablation on the coefficient $\gamma$.

| $\gamma$ | 0.4 | 0.5 | 0.6 | 0.7 | 0.8 | 0.9 |
|----------|-------|-------|-------|-------|-------|-------|
| 60M | 32.60 | 32.44 | 32.49 | 32.25 | 32.19 | 32.44 |
| 130M | 24.62 | 24.34 | 24.24 | 24.01 | 24.22 | 24.51 |

**Ablation on the complementary design choice.** To further validate our design choice of initializing the sparse module with complementary information from the remaining subspace, we conducted an analysis of the cosine similarity between the outputs of the low_rank and sparse components during training. We compared LOST against alternative initialization strategies on both 60M and 130M LLaMA models: using the same high-rank information as the low-rank module (High_rank) and using random rank information (Random_rank). The results14 demonstrate that LOST consistently achieves the lowest cosine similarity across all measured training steps, providing empirical evidence that our SVD-based complementary design maintains orthogonality better than alternative approaches throughout the entire training process. This persistent advantage validates that our initialization strategy guides the model toward maintaining complementary representations rather than redundant ones.

**Throughput comparison** LOST also provides computational advantages in terms of training and inference throughput. We evaluated the actual tokens processed per second on the 350M model, comparing LOST against its closest baseline SLTrain. As table15 shows that LOST achieves superior throughput in both training and inference scenarios.

## 5 CONCLUSION

In this paper, we presented LOST, a novel method for training low-rank LLMs from scratch through sparse plus low-rank decomposition. LOST effectively leverages the complementary relationship between sparse and low-rank components by decomposing weight matrices based on singular values, preserving both global structure in the low-rank component and essential local features in the sparse component. We demonstrated LOST's effectiveness through comprehensive experiments on the LLaMA models of different sizes (from 60M to 7b) trained on the C4 dataset. Our method achieved competitive performance while significantly reducing computational and memory requirements compared to traditional full-rank training approaches. Through detailed analyses of performance and memory efficiency across different model sizes, we validated the effectiveness of LOST and its practical benefits for efficient LLM training. We believe that this work makes significant progress in efficient LLM training and provides a solid foundation for relevant future research.

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

## A  APPENDIX

**Additional results for scaling performance** To further verify the performance of our method, we additionally trained a 7B full-rank model using the Adam optimizer. The training was performed on $16 \times$ NVIDIA H100 GPUs (80GB). Due to memory constraints with the full-rank model using Adam, we limited the batch size to 4, whereas other methods employed a batch size of 8. Given limited computational resources, we trained full-rank Adam model up to 40K steps with the same learning rate as LOST. Table 10 supplements Table 2 in the main text by reporting performance results of various methods for the 7B model at different training steps. At 40K steps, the Full-rank Adam model achieved a perplexity of 20.05, which remains higher than both LOST and 8-bit LOST at the same training step. We extended LOST training using Adam up to 150K steps, observing that LOST maintained stable convergence and consistently outperformed 8-bit Adam and 8-bit GaLore. In contrast, the full-rank Adam model experienced a sharp increase in evaluation loss around 12K steps, likely due to overfitting, further highlighting the instability of full-rank training with Adam and underscoring the robustness and scalability of LOST.

**The effectiveness of the structured sparsity** We compare channel-wise structured sparsity with element-wise unstructured sparsity to validate our design choice. Note that, LOST adopts activation functions in the low-rank matrices. As a result, the widely used method to merge the unstructured sparse component with low-rank matrices at the weight level is unfeasible. Therefore, we process the input through the unstructured matrix first, and then combine this output with the sparse output to ensure fair comparison with LOST. The perplexity and parameter count comparisons are presented in Table 11. We can see that the structured sparsity achieves better performance while requiring fewer parameters. The parameter difference stems from storage overhead: unstructured sparsity stores both

Table 10: Validation perplexity and actual memory footprint per GPU were reported for the LLaMA-7B model pre-trained on the C4 dataset for 40K steps. Baseline (except Full-Rank Adam) results are collected from Zhao et al. (2024); Han et al. (2024). LOST uses Adam optimizer and 8-bit LOST uses 8-bit Adam optimizer.

| Method | Batch size | Mem (G) | 10K | 40K | 80K | 120K | 150K |
|---|---|---|---|---|---|---|---|
| Full-Rank Adam | 4 | 49.53 | 24.95 | 20.05 | | N/A | |
| 8-bit Adam | 8 | 72.59 | N/A | 18.09 | 15.47 | 14.83 | 14.61 |
| 8-bit GaLore | 8 | 65.16 | 26.87 | 17.94 | 15.39 | 14.95 | 14.65 |
| 8-bit SLTrain | 8 | 60.91 | 27.59 | | N/A | | |
| LOST | 8 | 62.15 | **24.41** | **16.48** | **14.01** | **12.93** | **12.80** |
| 8-bit LOST | 8 | 50.19 | 24.67 | 17.59 | 15.16 | N/A | |

a full binary mask and the sparse weights in the original matrix shape, while structured sparsity only stores the selected channel indices and the corresponding channel weights.

**Limitation** The limitation of our current work is that we have only validated LOST on models up to 7B parameters, while many state-of-the-art LLMs in practical use today operate at scales of tens or even hundreds of billions of parameters. Due to computational resource constraints, we were unable to thoroughly evaluate our approach on these extremely large models. The scaling behavior of our low-rank and sparse decomposition method at such massive scales remains an open question. Investigating LOST's effectiveness and efficiency on models exceeding 10B parameters would be a valuable direction for future work.

**Broader impact** As LLMs become increasingly important across various applications, training and deploying such large-scale models has become standard practice in both industry and academia. However, training these massive models consumes substantial energy resources. Our work on LOST provides a promising approach to achieve comparable performance with significantly fewer parameters, offering a practical path toward more sustainable AI development. This could make LLMs more accessible to researchers and organizations with limited computational resources. We believe this work will inspire the broader research community to explore more efficient training methods.

Table 11: Perplexity and parameter count for structured and unstructured sparsity.

| Method | Perplexity | | Parameter Count | |
|---|---|---|---|---|
| | Structured | Unstructured | Structured | Unstructured |
| 60M | 32.25 | 34.18 | 43M | 68M |
| 130M | 24.01 | 26.35 | 94M | 178M |

Table 12: Hyperparameters of the LLaMA model. Training data is specified in tokens.

| Params | Hidden | Intermediate | Heads | Layers | Training Tokens |
|---|---|---|---|---|---|
| 60M | 512 | 1376 | 8 | 8 | 1.3B |
| 130M | 768 | 2048 | 12 | 12 | 2.6B |
| 350M | 1024 | 2736 | 16 | 24 | 6.4B |
| 1B | 2048 | 5461 | 24 | 32 | 13.1B |
| 7B | 4096 | 11008 | 32 | 32 | 19.7B |

## A.1 LLM USAGE

LLMs were used for grammar checking and language polishing to improve manuscript clarity.

Table 13: Hyperparameters of LOST for fine-tuning on GLUE datasets. We fix the epoch as 40. We apply the sparsity and coefficient as $\rho = 0.005$ and $\gamma = 0.7$ to all the experiments, respectively

|               | CoLA | STS-B | MRPC | RTE | SST-2 | MNLI | QNLI | QQP |
|---------------|------|-------|------|-----|-------|------|------|-----|
|               | Rank $r = 3$ | | | | | | | |
| Batch Size    | 64   | 16    | 16   | 16  | 16    | 16   | 16   | 16  |
| Learning Rate | 3e-4 | 3e-4  | 1e-4 | 5e-4| 3e-5  | 3e-4 | 5e-4 | 5e-5|
| LORA $\alpha$ | 12   | 24    | 6    | 6   | 6     | 6    | 12   | 12  |
|               | Rank $r = 7$ | | | | | | | |
| Batch Size    | 64   | 16    | 16   | 16  | 16    | 16   | 16   | 16  |
| Learning Rate | 3e-4 | 3e-4  | 1e-4 | 5e-4| 3e-5  | 3e-4 | 5e-4 | 5e-5|
| LORA $\alpha$ | 14   | 7     | 7    | 7   | 7     | 14   | 7    | 14  |

Table 14: Cosine similarity between low-rank and sparse component outputs during training on 60M and 130M LLaMA models. Lower values indicate better complementarity between components.

| Method | Training Steps | | | | | | | | |
|--------|--------|-----------|---------|-----------|---------|------------|----------|------------|----------|
|        | Step 0 | Step 2.5k | Step 5k | Step 7.5k | Step 10k | Step 12.5k | Step 15k | Step 17.5k | Step 20k |
| **60M Model** | | | | | | | | | |
| Low_rank (LOST) | 0.069 | 0.018 | 0.016 | 0.015 | 0.015 | - | - | - | - |
| High_rank        | 0.084 | 0.023 | 0.021 | 0.020 | 0.020 | - | - | - | - |
| Random_rank      | 0.073 | 0.021 | 0.018 | 0.016 | 0.016 | - | - | - | - |
| **130M Model** | | | | | | | | | |
| Low_rank (LOST) | 0.070 | 0.025 | 0.022 | 0.020 | 0.020 | 0.019 | 0.019 | 0.018 | 0.018 |
| High_rank        | 0.082 | 0.028 | 0.026 | 0.025 | 0.024 | 0.024 | 0.025 | 0.024 | 0.024 |
| Random_rank      | 0.077 | 0.026 | 0.023 | 0.023 | 0.022 | 0.021 | 0.020 | 0.020 | 0.020 |

Table 15: Training and inference throughput comparison between LOST and SLTrain on 350M LLaMA model. Results reported in tokens per second.

| Method | Training (tokens/s) | Inference (tokens/s) |
|--------|---------------------|----------------------|
| SLTrain | 10,373 | 20,258 |
| LOST    | **11,721** | **23,532** |