# OpenReview forum: "LOST: Low-rank and Sparse Pre-training for Large Language Models"
_ICLR.cc/2026/Conference — Submitted to ICLR 2026_

### Official Review · Reviewer_ZVHK · 2025-10-26

**Soundness:** 2
**Presentation:** 2
**Contribution:** 2
**Rating:** 4
**Confidence:** 4

**Summary:**

This paper introduces LOST (Low-rank and Sparse Training), a method for efficient large language model pre-training that combines low-rank factorization and structured channel-wise sparsity within each linear layer. The core idea is to perform an SVD of each weight matrix at initialization, use the top-r singular components for a low-rank representation, and utilize the residual singular subspace to select a small subset of channels forming a sparse complementary matrix. Experiments on LLaMA-style models ranging from 60M to 7B parameters demonstrate that LOST achieves comparable or better perplexity than full-rank and recent low-rank baselines.

**Strengths:**

1. Well-motivated decomposition: The proposed combination of low-rank and structured sparse components leverages complementary subspaces derived from SVD, offering a principled way to retain model expressivity while saving memory.
2. Extensive ablation studies: The paper carefully analyzes the effects of rank, sparsity, combination coefficient γ, nonlinearity placement, and initialization schemes, supporting the design choices and validating the method’s robustness under different configurations.
3. Practical efficiency gains: LOST achieves notable reductions in memory and computational costs with consistent or improved perplexity across several model sizes. The structured sparsity design is hardware-friendly compared to unstructured sparsity.

**Weaknesses:**

1. Using only SVD for initialization without modifying the training process raises questions about the effectiveness of training at larger scales and for longer durations.
2. The use of activation functions makes it difficult to merge parameter matrices in a manner similar to LoRA, which limits its application scenarios. Additionally, this method introduces more hyper parameters.
3. The differences shown in ablation for different Wl and Ws strategies are not significant, casting doubt on the necessity of SVD and sparse approaches.
4. Some comparisons rely on results from prior work, making it unclear whether all baselines share identical training data, training budgets, hyper parameters, and seeds.
5. The main text focuses exclusively on pretraining perplexity; downstream fine-tuning results are only mentioned in the appendix without quantitative comparisons, limiting understanding of real-world benefits.
6. The citation format contains widespread errors, with improper use of \citet and \citep commands.

**Questions:**

1. In channel-wise sparse weight part, what is the difference between column sparsity and row sparsity, and has this distinction been experimentally compared?
2. After using the activation function, why can we still do weight avg? (line 402)
3. Can the sparse part also be represented as a special low-rank form of AB, for example, where A is learnable and B is a sparse matrix containing only 0s and 1s to place the results in specific channels.
4. Why is the channel selected in the original matrix rather than in the W_comp in the method?
5. In the LOST implementation of Attention, are the QKV projection matrices processed separately, and are the heads also processed separately?
6. Have you tried the method on the MoE architecture?
7. Why not further apply SVD and sparse to adjust weights during training (similar to GaLore), but only use them during initialization?

---

> ### Author Response · Authors · 2025-11-23
> **Response to Reviewer ZVHK [1/3]**
>
> We sincerely thank the reviewer for providing thoughtful feedback and recognizing the strengths of LOST. We address your concerns and questions below.
>
> **W1:Using only SVD for initialization without modifying the training process raises questions about the effectiveness of training at larger scales and for longer durations.**
>
> The design of LOST introduces an activation function between the low-rank matrices to enhance representation capacity. A trade-off of this design is that it makes intermediate re-factorization difficult during training. We acknowledge this as a current limitation and plan to explore solutions that enable more flexible parameter updates during training.
>
> Regarding initialization, it is well established that initialization strongly influences model performance. LOST leverages a complementary SVD-based low-rank + structured sparse initialization, which empirically enhances representation capacity and consistently yields strong results across multiple model scales. In addition to the LLaMA experiments, we have now included results on T5-small, GPT-2 small, and GPT-2 medium, as referenced in our response to Reviewer h6tr, further demonstrating LOST’s robustness across different model.
>
> **W2:The use of activation functions makes it difficult to merge parameter matrices in a manner similar to LoRA, which limits its application scenarios. Additionally, this method introduces more hyper parameters.**
>
> We agree that the use of the activation function brings both benefits and drawbacks. Its main advantage is a substantial increase in representational capacity, which improves pre-training performance. The drawback is reduced compatibility with weight-merging techniques. For inference, while the weights cannot be merged into a single dense matrix, the sparse component is structured (column-wise), allowing for efficient inference. We are actively exploring ways to mitigate these limitations in future work.
>
> **W3:The differences shown in ablation for different Wl and Ws strategies are not significant, casting doubt on the necessity of SVD and sparse approaches.**
>
> To demonstrate the necessity of the proposed components, we present a direct comparison between Full-Rank, LOST, and a Low-Rank only baseline (initialized via SVD, without the sparse component) on 60M and 130M models:
> | method | Fullrank | LOST | Lowrank |
> | --- | --- | --- | --- |
> | 60M  | 34.06 | 32.25 |  32.93 |
> | 130M  | 24.36 | 24.01 |  24.74 |
>
> As shown, LOST consistently outperforms the Low-rank baseline (improving PPL by ~0.7). Furthermore, on the 130M model, LOST outperforms the Full-Rank baseline. This confirms that the co-design of low-rank and sparse components provides a benefit over using either component in isolation.
>
> **W4:Some comparisons rely on results from prior work, making it unclear whether all baselines share identical training data, training budgets, hyper parameters, and seeds.**
>
> We ensure fair comparisons following the experimental setup established in SLTrain [1], including data, training budgets, and random seeds. To ensure accuracy, we cited the reported results from baselines that utilized the same configurations. We also conducted extensive ablations to evaluate each component in LOST. We believe these experiments ensure that all comparisons are fair and our conclusions are reliable and reproducible.
>
> [1] SLTrain: a sparse plus low rank approach for parameter and memory efficient pretraining, NeurIPS 2024.

---

> ### Author Response · Authors · 2025-11-23
> **Response to Reviewer ZVHK [2/3]**
>
> **W5:The main text focuses exclusively on pretraining perplexity; downstream fine-tuning results are only mentioned in the appendix without quantitative comparisons, limiting understanding of real-world benefits.**
>
> Although LOST is designed for pre-training, we agree that demonstrating downstream generalizability is meaningful. Below are the experimental results on the GLUE benchmark using the hyperparameters provided in Table 13.
>
> | Method            | Memory | CoLA  | STS-B | MRPC  | RTE   | QNLI  | QQP   |
> |------------------|--------|-------|-------|-------|-------|-------|-------|
> | Full-size         | 747M  | 62.24 | 90.92 | 91.30 | 79.42 | 92.28 | 86.28 |
> |||||||||
> | LoRA, r = 4       | 257M  | 61.38 | 90.57 | 91.07 | 78.70 | 91.29 | 85.61 |
> | GaLore, r = 4     | 253M  | 60.35 | 90.73 | 92.25 | 79.42 | 91.06 | 85.89 |
> | LOST, r = 3       | 257M  | 60.96 | **90.88** | **93.15** | **79.54** | 90.86 | **86.04** |
> |||||||||
> | LoRA, r = 8       | 264M  | 61.83 | 90.80 | 91.90 | 79.06 | 91.22 | 85.93 |
> | GaLore, r = 8     | 257M  | 60.06 | 90.82 | 92.01 | 79.78 | 91.11 | 85.94 |
> | SLTrain, r = 8    |   -   | 60.35 | 90.74 | 92.38 | 79.42 | 91.27 | 85.93 |
> | LOST, r = 7       | 264M  | **62.52** | **91.15** | **93.68** | 79.68 | 91.17 | **86.51** |
>
> We compare LOST against LoRA, Galore, LORO, and SLTrain with $r=4$ and 8 to fine-tune the pre-trained RoBERTa-base model [1] on GLUE benchmark datasets. Notably, LOST uses slightly lower ranks ($r=3$ and $r=7$) to maintain comparable parameter counts after including the sparse component. Following the setup in [2], we modify the fine-tuning weight $W$ to $W + AB^T + W_s$ form, where $W$ represents the full-rank pre-trained weights, $AB^T$ is the low-rank module, and $W_s$ is the channel-wise structured sparse matrix.
>
> The resluts show that LOST achieves competitive or superior performance across GLUE tasks compared with baselines, validating its effectiveness and generalizability beyond the pre-training setting. Since fine-tuning typically involves much fewer parameters and training data compared to pre-training, standard LoRA is often sufficient for these tasks. This explains why low-rank methods show only marginal improvements over full-rank models or vanilla LoRA in fine-tuning scenarios. We would like to note that LOST is propsoed for efficient pre-training and this experiment aims to illustrate its generalization ability.
>
> [1]. Roberta: A robustly optimized bert pretraining approach, arXiv 2019.
> [2]. SLTrain: a sparse plus low rank approach for parameter and memory efficient pretraining, NeurIPS 2024.
>
> **W6:The citation format contains widespread errors, with improper use of \citet and \citep commands.**
>
> We appreciate the reviewer pointing this out. We will correct all citation formatting errors in the revised version.
>
> **Q1:In channel-wise sparse weight part, what is the difference between column sparsity and row sparsity, and has this distinction been experimentally compared?**
>
> We utilized column-wise sparsity primarily due to dimension compatibility in the forward pass. As shown in Eq. 4 , the output is computed as the sum of the low-rank component and the sparse component ($x_{[:,\mathcal{I}]}W_{s}^{T}$).
> Column-wise: Selecting columns from $W$ allows the input $x$ to be sliced ($x_{[:,\mathcal{I}]}$), resulting in a matrix multiplication that naturally matches the output dimension of the low-rank component.
> Row-wise: Selecting rows would alter the output feature dimension, causing the output shape of the sparse component to differ from that of the low-rank component. This prevents a direct summation without additional projection layers. While row-wise sparsity is an interesting direction, it would require a different architectural integration, which we leave for future work.
>
> **Q2:After using the activation function, why can we still do weight avg? (line 402)**
>
> In this ablation study, we did not include activation functions for either Weight Avg or Output Avg. Our goal here is to show that, without activation functions, Weight Avg and Output Avg achieve comparable performance. However, the advantage of Output Avg is that it allows us to insert nonlinear activation functions between the low-rank matrices, thereby further enhancing the model’s expressiveness. Although adding activation functions prevents the low-rank and sparse components from being merged back into a single model, the performance gains it brings make this trade-off worthwhile. We will make this point clear in the revised version of paper.

---

> ### Author Response · Authors · 2025-11-23
> **Response to Reviewer ZVHK [3/3]**
>
> **Q3:Can the sparse part also be represented as a special low-rank form of AB, for example, where A is learnable and B is a sparse matrix containing only 0s and 1s to place the results in specific channels.**
>
> This is an insightful suggestion. One could imagine a variant where the sparse matrix acts as a selector applied to a full matrix A. While this is outside the scope of the current paper, it is an interesting direction that we are considering for a potential follow-up. We thank the reviewer for this idea.
>
> **Q4:Why is the channel selected in the original matrix rather than in the W_comp in the method?**
>
> The design rationale is to minimize information loss. We calculate the importance score using the residual matrix $W_{comp}$ to identify which channels to maintain. Once indices are identified, we copy the actual weights from the original Kaiming-initialized matrix $W$.
>
> If we took the weights directly from $W_{comp}$, we would be using values that have already suffered from SVD truncation error. By fetching the columns from the original $W$, we preserve the precision of those specific important channels.
>
> To validate this, we add two more experiments on 60m and 130m Llama model which use the same configuration with Table 1 in the main paper but with the column-wise sparse components selected based on $W_{comp}$. The results are shown below:
>
> | method | $W_{original}$ | $W_{comp}$ |
> | --- | --- | --- |
> | 60M  | 32.25 | 32.42 |
> | 130M  | 24.01 | 24.22 |
>
> The results shows that initializing the column-wise sparse components based on $W_{original}$ lead to consistent performance gain on both 60m and 130m model, which validate our method.
>
> **Q5:In the LOST implementation of Attention, are the QKV projection matrices processed separately, and are the heads also processed separately?**
>
> Yes. LOST is applied to all linear layers inside the attention/MLP blocks but not the lm_head layer.
>
> **Q6:Have you tried the method on the MoE architecture?**
>
> Not yet. Applying LOST to other architectures (such as MoE and diffusion models) is a promising future direction. We plan to extend LOST to these architectures in the future work.
>
> **Q7:Why not further apply SVD and sparse to adjust weights during training (similar to GaLore), but only use them during initialization?**
>
> As mentioned in W1, the primary reason is the use of the activation function within the low-rank module. This non-linearity prevents the straightforward merging and re-factorization of weights during training. We prioritized the performance gains from the activation function for this work, but enabling dynamic updates alongside non-linearity is a challenge we aim to address in the future.

---

> ### Author Response · Authors · 2025-11-27
>
> Dear Reviewer ZVHK,
>
> We sincerely appreciate your insightful review, which has been helped in enhancing the quality of our work. As we approach the end of the discussion phase, please don't hesitate to let us know if you have any further concerns, and we would be more than happy to address them.
>
> Best, Authors

---

### Official Review · Reviewer_h6tr · 2025-10-30

**Soundness:** 3
**Presentation:** 3
**Contribution:** 3
**Rating:** 6
**Confidence:** 3

**Summary:**

This paper proposes a novel method called LOST, which aims to address the massive computational and memory overhead challenges encountered when pre-training LLMs from scratch. The authors cleverly combine low-rank and sparse structures, with its core innovation lying in "co-design" via SVD: the largest singular values are used to construct the low-rank component for capturing key information, while the remaining singular values are leveraged to build a channel-wise sparse component to compensate for the information lost in low-rank approximation. Experimental results demonstrate that across multiple model scales, LOST can significantly reduce resource consumption while achieving performance that equals or even surpasses that of full-rank models.

**Strengths:**

1. The paper is well written and easy to read.
2. Significant efficiency and performance advantages. The core results (Table 1, Figure 1) show that while drastically reducing memory footprint, LOST achieves lower Perplexity than full-rank models across most model scales, demonstrating the notable superiority of the proposed method.
3. The experiments cover a variety of model sizes ranging from 60M to 7B parameters, verifying the method’s universality and scalability. Comparisons are conducted with Full-Rank, LoRA, and a number of state-of-the-art low-rank pre-training methods (GaLore, LORO, CoLA, SLTrain), and the results show that LOST outperforms many baseline methods.
4. The authors have conducted in-depth ablation studies on nearly every design detail of the model.

**Weaknesses:**

1. There are only experimental results of pre-training on LLMs, with a lack of fine-tuning results on downstream tasks (e.g., GLUE). The fine-tuning hyperparameters for the GLUE dataset are provided in Table 13; however, the actual experimental results of the GLUE fine-tuning task are not presented in the main text.
2. The authors only conducted experiments on models from the Llama family, and there is a lack of experimental results on LLMs with other architectures. It remains unclear whether LOST is also effective for other models.
3. In Table 1, the PPL of LOST is significantly superior to that of the full-rank model on the small 60M-parameter model, but only slightly better on the 1B-parameter model. As the model scale increases, the advantage of LOST over the Full-Rank model becomes smaller.

**Questions:**

1. How is the "actual memory footprint" in Table 2 measured? Can the authors provide more detailed specifics?

---

> ### Author Response · Authors · 2025-11-23
> **Response to Reviewer h6tr [1/2]**
>
> We sincerely thank the reviewer for the effort spent assessing our work. We are encouraged that you recognize the novelty of LOST’s co-design framework and the experiments we conducted. We address your comments and questions below.
>
> **W1:There are only experimental results of pre-training on LLMs, with a lack of fine-tuning results on downstream tasks (e.g., GLUE).**
>
> Although LOST is designed for pre-training, we agree that demonstrating downstream generalizability is meaningful. Below are the experimental results on the GLUE benchmark using the hyperparameters provided in Table 13.
>
> | Method            | Memory | CoLA  | STS-B | MRPC  | RTE   | QNLI  | QQP   |
> |------------------|--------|-------|-------|-------|-------|-------|-------|
> | Full-size         | 747M  | 62.24 | 90.92 | 91.30 | 79.42 | 92.28 | 86.28 |
> |||||||||
> | LoRA, r = 4       | 257M  | 61.38 | 90.57 | 91.07 | 78.70 | 91.29 | 85.61 |
> | GaLore, r = 4     | 253M  | 60.35 | 90.73 | 92.25 | 79.42 | 91.06 | 85.89 |
> | LOST, r = 3       | 257M  | 60.96 | **90.88** | **93.15** | **79.54** | 90.86 | **86.04** |
> |||||||||
> | LoRA, r = 8       | 264M  | 61.83 | 90.80 | 91.90 | 79.06 | 91.22 | 85.93 |
> | GaLore, r = 8     | 257M  | 60.06 | 90.82 | 92.01 | 79.78 | 91.11 | 85.94 |
> | SLTrain, r = 8    |   -   | 60.35 | 90.74 | 92.38 | 79.42 | 91.27 | 85.93 |
> | LOST, r = 7       | 264M  | **62.52** | **91.15** | **93.68** | 79.68 | 91.17 | **86.51** |
>
>
> We compare LOST against LoRA, Galore, LORO, and SLTrain with $r=4$ and 8 to fine-tune the pre-trained RoBERTa-base model [1] on GLUE benchmark datasets. Notably, LOST uses slightly lower ranks ($r=3$ and $r=7$) to maintain comparable parameter counts after including the sparse component. Following the setup in [2], we modify the fine-tuning weight $W$ to $W + AB^T + W_s$ form, where $W$ represents the full-rank pre-trained weights, $AB^T$ is the low-rank module, and $W_s$ is the channel-wise structured sparse matrix.
>
> The resluts show that LOST achieves competitive or superior performance across GLUE tasks compared with baselines, validating its effectiveness and generalizability beyond the pre-training setting. Since fine-tuning typically involves much fewer parameters and training data compared to pre-training, standard LoRA is often sufficient for these tasks. This explains why low-rank methods show only marginal improvements over full-rank models or vanilla LoRA in fine-tuning scenarios. We would like to note that LOST is propsoed for efficient pre-training and this experiment aims to illustrate its generalization ability.
>
> [1]. Roberta: A robustly optimized bert pretraining approach, arXiv 2019.
> [2]. SLTrain: a sparse plus low rank approach for parameter and memory efficient pretraining, NeurIPS 2024.
>
> **W2:The authors only conducted experiments on models from the Llama family.**
>
> We thank the reviewer for this constructive suggestion. To further examine LOST's generality beyond the LLaMA architecture, we conducted additional pre-training experiments on T5-small, GPT-2 small, and GPT-2 medium using 1.1B, 2.2B, and 6.4B training tokens respectively. We compared LOST against full-rank training and SLTrain. All methods use the same learning rate (0.001), and the ranks for LOST and SLTrain are matched to ensure equal trainable parameter counts.
>
> T5-small:
>
> |Method	|Full rank	|LOST|	SLTrain	|
> |---------------------|--------|-------|-------|
> |Evaluation loss|	5.67	|7.74	|9.15	|
>
> GPT2-small:
>
> |Method	|Full rank	|LOST|	SLTrain	|
> |---------------------|--------|-------|-------|
> |Evaluation loss|	3.42	|3.57|3.96|
>
> GPT2-medium:
> |Method	|Full rank	|LOST|	SLTrain	|
> |---------------------|--------|-------|-------|
> |Evaluation loss|	3.26	|3.23 |	3.65 |
>
> These results show that LOST consistently outperforms SLTrain and shows competitive or even better performance compared with full-rank models. These observations demonstrate that LOST's effectiveness generalizes beyond the LLaMA family to diverse transformer architectures.

---

> ### Author Response · Authors · 2025-11-23
> **Response to Reviewer h6tr [2/2]**
>
> **W3:In Table 1, the PPL of LOST is significantly superior to that of the full-rank model on the small 60M-parameter model, but only slightly better on the 1B-parameter model. As the model scale increases, the advantage of LOST over the Full-Rank model becomes smaller.**
>
> We appreciate this insightful observation. This trend reflects two well-established phenomena in efficient model training. The first is the scaling challenge. It is widely observed that it is difficult for the low-rank model to match the full-rank performance as the model size scales [1,2,3]. This is a fundamental challenge in the field rather than a limitation specific to LOST. The second is the diminishing marginal gains. Larger models achieve much lower perplexity, making further reductions inherently more difficult.
>
> More importantly, LOST's primary contribution is not outperforming full-rank models, but rather achieving competitive performance while reducing computational and memory costs. We will explore techniques such as periodically resetting parameters through SVD-based decomposition to further enhance performance in future work.
>
> **Q1:How is the "actual memory footprint" in Table 2 measured? Can the authors provide more detailed specifics?**
>
> We measured the maximum memory allocation [2] during the training using the PyTorch code torch.cuda.max_memory_allocated().
>
> [1]. GaLore: Memory-Efficient LLM Training by Gradient Low-Rank Projection, ICML 2024.
> [2]. CoLA: Compute-Efficient Pre-Training of LLMs via Low-Rank Activation, EMNLP 2025.
> [3]. SLTrain: a sparse plus low-rank approach for parameter and memory efficient pretraining, NeurIPS 2024.

---

> ### Author Response · Authors · 2025-11-27
>
> Dear Reviewer h6tr,
>
> We are truly thankful for your insightful feedback, which has significantly enhanced our work. As we approach the conclusion of the discussion phase, we would be happy to answer if you have any more concerns.
>
> Best, Authors

---

### Official Review · Reviewer_xG1E · 2025-10-31

**Soundness:** 2
**Presentation:** 3
**Contribution:** 2
**Rating:** 2
**Confidence:** 4

**Summary:**

This paper introduces LOST (LOw-rank and Sparse pre-Training), a new method to efficiently pre-train large language models from scratch. Its key idea is to "co-design" low-rank and sparse components so they are complementary. It initializes by performing a Singular Value Decomposition (SVD) on the full-rank weight matrices. The dominant singular values are used to create the main low-rank component. The remaining, smaller singular values are then used to construct a channel-wise sparse component that captures information lost in the low-rank truncation.

The method was evaluated on pre-training LLaMA models from 60M to 7B parameters. Experiments show that LOST achieves perplexity that is competitive with or even superior to full-rank models, while significantly reducing memory and computational overhead.

**Strengths:**

1. Principled "Co-Design" Methodology: Instead of naively combining low-rank and sparse matrices, it uses Singular Value Decomposition (SVD) to purposefully create the sparse component from the residual singular values. This "co-design" ensures the two components are complementary from the start.
2. Strong Empirical Performance: LOST demonstrates state-of-the-art results. It achieves perplexity scores that are competitive with, or even superior to (very suspicious they use different batch size and train steps in Table 10.) , standard full-rank pre-training on models up to 1B parameters. Furthermore, it clearly outperforms direct competitors in efficient pre-training, such as SLTrain, LORO, and GaLore, in like-for-like comparisons.

**Weaknesses:**

(Please respond to the questions section)

1. Limited Novelty Compared to SLTrain
2. Weak Justification for Sparse Component Design
3. Inconsistent and Unfair Experimental Comparisons

**Questions:**

This paper's core claims are undermined by concerns regarding limited novelty, weak justification for its methodology, and several inconsistencies in the experimental results.
1. The paper's novelty appears incremental. The proposed framework, which combines a low-rank component with a sparse matrix defined by a predefined, static mask(column-wise), is functionally identical to the SLTrain method. The only significant difference is the initialization strategy for this sparse mask (using SVD residuals) and the low-rank factors. This raises the question of whether LOST is a new training method or, more accurately, a new initialization technique for an existing one.
2. The paper's "co-design" hinges on creating the sparse component from the L2-norm of the columns of the residual matrix $W_{comp}$. First, the paper provides no strong theoretical or empirical justification for why this specific choice (column-wise L2-norm) is optimal over, for example, a row-wise selection or an L1-norm; Second, the ablation study in Table 3, which attempts to justify this, shows results that are not significant. For the 60M model, the perplexity for the proposed $SVD_{l2}^{rem}$ is 32.25, while $SVD_{l1}^{rem}$ is 32.33. A difference of 0.08 PPL is well within the standard deviation of such experiments and fails to convincingly argue for the superiority of this specific design choice.
3. Several experimental results appear to be "not normal" and suffer from confounding variables or internal contradictions, making the paper's conclusions difficult to trust. 1. Unfair 7B Model Comparison: In Table 10, the 7B LLaMA experiment compares LOST (trained with a batch size of 8) to a Full-Rank Adam baseline (trained with a batch size of 4). Batch size is a critical hyperparameter, and this discrepancy makes the comparison unfair and invalid. The Full-Rank model is severely disadvantaged, and no reliable conclusion can be drawn; 2. Contradictory $\gamma$ Ablation: This is the most serious concern. The $\gamma$ parameter balances the low-rank and sparse components. As $\gamma$ approaches 1, the sparse component's contribution $(1-\gamma) \cdot x_{[:,\mathcal{I}]}W_{s}^{T}$ should vanish, and LOST's performance should converge to that of a LoRA-style model (with SVD initialization). But the results contradict this. For the 130M model, LOST with $\gamma=0.9$ (mostly low-rank) achieves a perplexity of 24.51 (Table 9), however, the baseline LoRA model in Table 1 reports a perplexity of 33.92. This is a massive discrepancy. It strongly suggests that the "LoRA" baseline in Table 1 was trained with a standard (e.g., Kaiming) initialization, while the "LOST" model in Table 9 benefits from SVD initialization. This implies that the entire performance gain may be coming from the SVD initialization alone, not the novel sparse+low-rank co-design. This experiment fails to isolate the variable it claims to be testing.

In conclusion, I would not recommend accepting this work given the current quality and concerns.

---

> ### Author Response · Authors · 2025-11-23
> **Response to Reviewer xG1E [1/2]**
>
> We thank the reviewer for highlighting areas that required further clarification. We address your concerns as below.
>
> **Q1:The paper's novelty appears incremental.**
>
> We respectfully disagree that our work lacks novelty. Innovation does not necessarily require inventing a completely new paradigm from scratch. As it stated in the ICLR 2026 Reviewer Guide [1], strong papers often provide technically sound, reproducible, and novel findings that improve upon prior approaches. LOST makes several fundamental advances over SLTrain and other related baselines:
> 1. Complementary SVD-based co-initialization of the low-rank and sparse components, which substantially improves pre-training stability and performance.
> 2. Column-wise structured sparsity design, which not only reduces parameter count but also enables compute-efficient pre-training—something SLTrain does not provide.
> 3. A nonlinear activation between the low-rank factors, which significantly enhances representation capacity and fundamentally changes the behavior of the low-rank module compared to prior linear decompositions.
>
> These elements produce a framework that is substantially different from SLTrain in both design and empirical behaviour. We hope this could clarify LOST’s contribution.
>
> **Q2:The paper's "co-design" hinges on creating the sparse component from the L2-norm of the columns of the residual matrix. First, the paper provides no strong theoretical or empirical justification for why this specific choice (column-wise L2-norm) is optimal; Second, the ablation study in Table 3 shows results that are not significant. For the 60M model, the perplexity for the proposed is 32.25, while is 32.33. A difference of 0.08 PPL is well within the standard deviation of such experiments and fails to convincingly argue for the superiority of this specific design choice.**
>
> We would like to clarify that one of our key innovation regarding sparsity is not the L2 norm itself, but the design of a complementary initialization strategy where the sparse component captures information not represented by the principal SVD ranks. This is a core conceptual difference from SLTrain and other baselines.
> Regarding the choice of L1/L2 norms: Both are valid and were explored — in practice, both of them outperform using only principal rank with low-rank model or random initialization as in SLTrain. The specific norm is not the source of LOST's improvement but the complementary co-design is.
> Regarding why row-wise selection was not used: As shown in Eq. (4) in the manuscript, the output is computed as input × W_t, meaning that the column-wise structure is necessary for the sparse component to produce an output of the same shape as the low-rank component. Row-wise selection would break this compatibility and prevent natural combination of the two components. We acknowledge the reviewer’s suggestion and will consider exploring alternative structured methods in future work.
>
> [1] ICLR 2026 Reviewer Guide, 2025.

---

> ### Author Response · Authors · 2025-11-23
> **Response to Reviewer xG1E [2/2]**
>
> **Q3:Several experimental results appear to be "not normal" and suffer from confounding variables or internal contradictions, making the paper's conclusions difficult to trust. 1. Unfair 7B Model Comparison: In Table 10, the 7B LLaMA experiment compares LOST (trained with a batch size of 8) to a Full-Rank Adam baseline (trained with a batch size of 4). Batch size is a critical hyperparameter, and this discrepancy makes the comparison unfair and invalid. The Full-Rank model is severely disadvantaged, and no reliable conclusion can be drawn; 2. Contradictory Ablation: This is the most serious concern. The parameter balances the low-rank and sparse components. As approaches 1, the sparse component's contribution should vanish, and LOST's performance should converge to that of a LoRA-style model (with SVD initialization). But the results contradict this. For the 130M model, LOST with (mostly low-rank) achieves a perplexity of 24.51 (Table 9), however, the baseline LoRA model in Table 1 reports a perplexity of 33.92. This is a massive discrepancy. It strongly suggests that the "LoRA" baseline in Table 1 was trained with a standard (e.g., Kaiming) initialization, while the "LOST" model in Table 9 benefits from SVD initialization. This implies that the entire performance gain may be coming from the SVD initialization alone, not the novel sparse+low-rank co-design. This experiment fails to isolate the variable it claims to be testing.**
>
> We would like to clarify that the full-rank 7B model could only be trained with batch size 4 due to memory limitations on 80GB GPUs, resulting in the different batch size settings in Table 10. To address this concern and ensure a fair comparison,  we used 8 B200 GPUs with larger memory capacity and retrained the Full-Rank baseline with batch size 8 and learning rate 0.001, exactly matching LOST's settings. Note that, due to computational budget limitations, we were able to train for 6K steps. The updated results are presented below:
>
> | Method | Steps | PPL |
> | --- |  --- | --- |
> | Full-rank| 6k | 49.99 |
> | LOST | 6k |28.67 |
>
> The results demonstrate that at 6K steps, LOST achieves substantially better performance (28.67 vs 49.99 PPL), indicating faster convergence. We note that the full-rank model's poor performance at learning rate 0.001 may reflect its need for more careful hyperparameter tuning, as full-rank models often require smaller learning rates for stable optimization. Although the model is still undertrained at this point, combining this observation with Table 10 in the main paper which shows LOST consistently outperforms other baseline methods indicates that LOST maintains strong pretraining performance when scaled up to 7B models.
>
> Regarding the "contradictory ablation", we respectfully clarify that this concern stems from a misunderstanding of what different experimental configurations represent. The LoRA baseline in Table 1 is taken from [2,3], where it contains a frozen W after initialization and training is performed only on the low-rank update (with the effective weights is the form of $W + AB$). This baseline is not equivalent to training with only low-rank models. The reviewer notes that if $\gamma=1.0$, the model becomes a pure low-rank model initialized via SVD. We did report this result in the paper: it corresponds to Sparsity=0 in Table 8. For clarity, we have combined these results into an expanded version of Table 9 below, explicitly showing the $\gamma=1.0$ case:
> | γ | 0.4 | 0.5 | 0.6 | 0.7 | 0.8 | 0.9 | 1.0 |
> | --- | --- | --- | --- | --- | --- | --- | --- |
> | 60M  | 32.60 | 32.44 | 32.49 | 32.25 | 32.19 | 32.44 | 32.93 |
> | 130M  | 24.62 | 24.34 | 24.24 | 24.01 | 24.22 | 24.51 | 24.74 |
>
> We can see that the SVD-based lowrank model ($\gamma=1.0$) achieves a perplexity of 24.74 (for 130M), whereas LOST ($\gamma=0.7$) achieves 24.01. This improvement confirms that the performance gain is not solely due to SVD initialization, but rather the specific combination of low-rank and sparse components.
>
> We hope these clarifications address your concerns. We sincerely hope you will reevaluate our work.
>
> [2] GaLore: Memory-Efficient LLM Training by Gradient Low-Rank Projection, ICML 2024.
> [3]. SLTrain: a sparse plus low rank approach for parameter and memory efficient pretraining, NeurIPS 2024.

---

> > ### Author Response · Authors · 2025-11-27
> > **Follow-up on Q3: Detailed component analysis on LoRA Baseline**
> >
> > It is important to note that all methods in the original ablation table with different $\gamma$ include the activation function introduced in CoLA. To clearly isolate the impact of different components on model performance, we conducted additional experiments on the 60M and 130M models using: (1) SVD-based initialization without the activation function; (2) Kaiming initialization for both A and B. The learning rates used in these experiments (except the LoRA baseline, which is directly taken from [2]) were selected via grid search, with candidate values {5e-4, 1e-3, 2e-3, 3e-3, 6e-3}.
> >
> > The full comparison is shown in the table below:
> > | methods | 60m | 130m |
> > | --- | --- | --- |
> > | LoRA  | 34.99 | 33.92 |
> > | Lowrank  | 43.01 | 36.46 |
> > | Lowrank + SVD  | 38.76 | 27.37 |
> > | Lowrank + SVD + activation | 32.93 | 24.74 |
> > | LOST | 32.25 | 24.01 |
> >
> > where:
> > - LoRA: Refers to the "W + AB" form, where W is frozen after initialization [2].
> > - Low-rank: Denotes a pure low-rank model with A and B initialized using Kaiming initialization.
> > - Low-rank + SVD: Replaces Kaiming initialization with SVD-based initialization.
> > - Low-rank + SVD + activation: Further adds the CoLA-style activation.
> >
> > These results clearly show the performance gains brought by LOST and the degradation that occurs when specific modules are removed. Although absolute improvements naturally diminish as the models reach lower perplexity values, the relative contribution of each component remains significant. We hope this table and analysis fully address your question regarding the necessity and effectiveness of the components in LOST.

---

> ### Author Response · Authors · 2025-11-27
>
> Dear Reviewer xG1E,
>
> We sincerely appreciate your thoughtful feedback, which has greatly contributed to improving our work. As we approach the conclusion of the discussion phase, please feel free to share any additional concerns. We would be more than happy to address them.
>
> Best, Authors

---

> > ### Comment · Reviewer_xG1E · 2025-11-27
> >
> > I thank the authors for their rebuttal. I believe the ablation table in the rebuttal is very helpful, showing the contribution of SVD and activation separately.
> >
> > I think my concern about $\gamma$-ablation is mitigated now (although I'm still not sure if this analysis could be carried to larger models). I still have some concern about the 7B experiment since the table in the rebuttal is for 40K steps, which means that the total tokens is far less than Chinchilla optimal and less representative.
> >
> > Conceptually, my biggest concern is that, based on my past experience on pre-training, the difference in initialization will be less significant after training for more steps (say 2x Chinchilla law, for 1b model it would be 40B tokens). I understand that it might be too tight in time to do more experiments. But could the author comment on this concern of mine?

---

> > > ### Author Response · Authors · 2025-11-28
> > >
> > > Dear Reviewer xG1E,
> > >
> > > Thank you for acknowledging our response regarding the $\gamma$ ablation. Regarding your concern about the impact of initialization over long-term training, we agree that using more data could potentially compensate for initialization issues. However, we would like to emphasize that our method is fundamentally designed for efficient training, which inherently includes data efficiency.  The core objective of efficient pre-training research is to achieve a good performance within a fixed compute budget.
> > >
> > > More importantly, prior work consistently highlights the importance of initialization for reducing training cost. For instance, [1] shows that optimal initialization variance depends on model depth, and inappropriate variance can harm performance even with more data. [2,3] studies how to adapt hyperparameters from small-scale model to large-scale model to reduce the training cost. One of the important facts is to adjust the layer-wise variance. [4] demonstrates that suboptimal initialization can lead to instability training. Widely used frameworks like deepspeed also emphasize the precision of initialization to prevent training divergence, further validating that initialization remains critical even for long-term training.
> > >
> > > To further address your concerns, we are currently running the 2×chinchilla experiments as you suggested. However, due to time and computational constraints during the rebuttal period, these experiments are not yet complete. We will include these results in the revised version. We believe our current results across multiple scales and evaluation tasks already provide strong evidence for LOST's efficiency advantages. Please consider reevaluating our work if we adequately addressed your concerns. We are happy to answer any further questions.
> > >
> > > [1] Mean Field Residual Networks: On the Edge of Chaos, NeurIPS 2024.
> > > [2] Feature Learning in Infinite-Width Neural Networks, ICML 2021.
> > > [3] Tuning Large Neural Networks via Zero-Shot Hyperparameter Transfer, NeurIPS 2021.
> > > [4] GradInit: Learning to Initialize Neural Networks for Stable and Efficient Training, NeurIPS 2021.
> > >
> > > Best, Authors

---

> > > > ### Author Response · Authors · 2025-12-03
> > > > **Follow-up on initialization efficiency ablation**
> > > >
> > > > Dear Reviewer xG1E,
> > > >
> > > > As promised, we conducted an additional ablation experiment on the 60M model trained for up to about 7x Chinchilla law data (8.4b for 60m model) to evaluate the impact of initialization with more training data. We compared the standard low-rank model with SVD-based initialization against a lowrank model using Kaiming initialization. We kept all other hyperparameters identical. The validation PPL at different tokens is reported below:
> > > >
> > > > | Method | 1x Chinchilla law | 2x |3x |4x | 5x |6x |7x |
> > > > | --- |  --- | --- |---|--- | --- |---|--- |
> > > > | Lowrank (SVD) | 37.82 | 35.21 |33.99| 33.27 |32.77|32.38|32.03|
> > > > | Lowrank (Kaiming) | 42.9 |39.04 |37.31 |36.31| 35.81 |35.22|34.83|
> > > >
> > > > The results show that the model using Kaiming initialization consistently underperforms compared to the SVD-initialized model. Even after converging in the later stages, the Kaiming-initialized model fails to match the performance of the SVD-initialized model. This shows that initialization is critical and cannot be simply compensated for by increasing training data.
> > > >
> > > > Best, Authors

---

### Official Review · Reviewer_a9CA · 2025-11-01

**Soundness:** 2
**Presentation:** 3
**Contribution:** 2
**Rating:** 4
**Confidence:** 4

**Summary:**

This work presents a new pre-training strategy that takes low-rank and channel-wise sparse matrices as the model weights. A low-rank plus sparse matrix is a typical solution for approximating complicated matrices, such as the parameter matrix in a neural network. The method aims to retain full-rank expressivity while reducing parameters and memory. Experiments from 60M to 7B models on a classical pre-training corpus C4 show that the proposed methods outperform baselines with notable memory savings.

**Strengths:**

Many Previous works have shown empirical results that large models exhibit substantial parameter redundancy. The success of LoRA also demonstrates that tuning the low-rank part of the weight matrix can effectively learn new things. Extending this idea to the pre-training is promising and could have a broad impact on the large model community.

**Weaknesses:**

1. My major concern is the lack of a scaling law. When comparing architectures/optimization paradigms, scaling laws are a crucial and widely used tool. Different methods and settings often require different optimal hyperparameters, and relative rankings can flip as scale changes (e.g., attention variants such as MLA, GQA, linear attention exhibit size-dependent crossovers). Although this work reports results of multiple model sizes, the comparisons could still be confounded by hyperparameter suboptimality at each scale.
2. It's better to analyze why the proposed method can beat the full-rank baseline in perplexity. In principle, a constrained parameterization should not outperform a full-rank baseline. I doubt the reason is that the constrained model regularizes better under non-optimal hyperparameters, early stopping, or limited training data. If the proposed method consistently outperforms the full-rank baseline, I think the author should provide more insights.
3. The amount of pre-training tokens seems to be small relative to standard LLM pre-training. I do not mean training a model at trillions of tokens like the popular open-sourced model, but I doubt the loss curve has entered a plateau or a very slow region, making the observation biased. Conclusions based on limited training data may not extrapolate. Constrained models sometimes look strong in the undertrained regime but fall behind when trained to convergence. Looking at the scaling law at the step level, such as the L(S) estimation, could be a good way to analyze the proposed method.
4. A minor issue: C4 is somehow out-of-date, conducting experiments with nemontron-cc or fineweb-edu could be a better choice.

**Questions:**

see weakness

---

> ### Author Response · Authors · 2025-11-23
> **Response to Reviewer a9CA**
>
> We sincerely thank the reviewer for the constructive feedback and the recognition of our work's potential. We have addressed your concerns as below.
>
> **W1:My major concern is the lack of a scaling law.**
>
> The primary reason we utilized the training token settings in Table 1 of the main paper was to ensure a fair comparison with existing baselines. To address the concern regarding scaling laws, we are retraining the 1B parameter model (both LOST and Full-Rank) using 20B tokens, adhering to the Chinchilla scaling laws (about 20 tokens per parameter). This experiment ensures the models are not under-trained. The results will be uploaded once its completed.
>
> **W2:Analyze why the proposed method can beat the full-rank baseline in perplexity.**
>
> Thank you for your insightsful observation. We provide the following insights supported by literatures to explain why LOST achieves competitive results to the full-rank model:
> 1. Existed works indicate that the weight matrices and gradients of Large Language Models naturally exhibit low-rank property during training [1, 2, 3]. This suggests that full-rank weights contain redundancy and pretraining from a low-rank model can in principle reach performance comparable to a full-rank model.
> 2. As noted in [4], the initialization variance of different network layers should be related to the network depth. Standard methods such as Kaiming initialization did not consider to adjust different variance to the network in the different layer. LOST’s co-design initialization strategy may implicitly adjust the variance thus achieve a better initialization.
> 3. Prior work suggests that relying solely on low-rank or sparse structures can harm performance [5, 6]. LOST addresses this by design the low-rank component to capture global features, while the sparse residual captures critical local features. This design does not restrict the model to only learn within a low-rank subspace. It prevents performance bottlenecks caused by limited parameter capacity.
> 4. LOST inserts an activation function between the low-rank matrices. As demonstrated in CoLA [7], this enhances the representation capacity of the layers beyond standard low-rank matrices.
>
> These factors provide a coherent explanation for why LOST could achieve competitive results with full-rank model.
>
> **W3:The amount of pre-training tokens seems to be small relative to standard LLM pre-training.**
>
> As noted in our response to W1, we conducted additional experiments on the 1B model using 20B tokens, consistent with the Chinchilla recommendation of ~20× data/parameter ratio. The results will be uploaded once its completed.
>
> **W4:Conducting experiments with nemontron-cc or fineweb-edu.**
>
> To address this, we have conducted new experiments using the FineWeb-Edu dataset. We trained the 60M and 130M Llama models using LOST, Full-Rank, and our closest baseline, SLTrain. We use the same hyperparameter setting as the C4 experiments. The results (PPL) are showed below:
>
> | method | Full-rank | SLTrain | LOST |
> | --- | --- | --- | --- |
> | 60M  | 20.01 | 24.49 |22.11 |
> | 130M  | 17.16 | 21.04 | 18.48|
>
> We can see that LOST demonstrates strong generalization ability on the FineWeb-Edu dataset, outperforming SLTrain and achieving performance close to that of the full-rank model.
>
> [1]. How many degrees of freedom do we need to train deep networks: a loss landscape perspective, ICLR 2022.
>
> [2]. GaLore: Memory-Efficient LLM Training by Gradient Low-Rank Projection, ICML 2024.
>
> [3]. From Low Rank Gradient Subspace Stabilization to Low-Rank Weights: Observations, Theories, and Applications, ICML 2025.
>
> [4]. Mean Field Residual Networks: On the Edge of Chaos, NeurIPS 2024.
>
> [5]. Initialization and regularization of factorized neural layers,  ICLR 2021.
>
> [6]. Exploring low rank training of deep neural networks, arXiv 2022.
>
> [7]. CoLA: Compute-Efficient Pre-Training of LLMs via Low-Rank Activation, EMNLP 2025.

---

> > ### Author Response · Authors · 2025-12-03
> >
> > Dear Reviewer a9CA,
> >
> > As promised, we provide the PPL comparison for the 1B model trained on 20B tokens. The full-rank baseline in Table 1 of the main paper utilized the grid-searched learning rate from [1]. Due to computational constraints, we trained both the Full-Rank model and LOST using the same learning rate of 2e-3.
> >
> > Results: The Full-Rank model achieved a ppl of 15.53, while LOST achieved 14.49.
> >
> > The results demonstrate LOST's consistent superior performance even with increased training data. We acknowledge that the full-rank model may achieve better results with extensive hyperparameter tuning. Taken together with our earlier results, this additional experiment further confirms LOST's performance.
> >
> > [1]. GaLore: Memory-Efficient LLM Training by Gradient Low-Rank Projection, ICML 2024.
> >
> > Best, Authors

---

> ### Author Response · Authors · 2025-11-27
>
> Dear Reviewer a9CA,
>
> We are truly grateful for your thoughtful comments, which has significantly contributed to the improvement of our work. If there are any additional feedback you‘d like us to consider, please let us know. Thank you for taking time and effort in reviewing our work.
>
> Best, Authors

---

### Author Response · Authors · 2025-12-03
**General Response**

Dear Area Chair,

Thank you for your time and effort for reviewing LOST. We also appreciate the constructive feedback from all reviewers, which has significantly strengthened our work and allowed us to further clarify our novelty and contribution behind LOST.

**Summary of Current Status**: Our scores are 6, 4, 4, and 2, with reviewers recognizing our novelty, contribution, and experimental improvements such as "core innovation lying in 'co-design' via SVD", "well-motivated decomposition", "strong empirical performance", and "notable memory savings". Importantly, **reviewer xG1E who gave the lowest score 2** engaged with us during the discussion phase, and agreed that **almost all the concerns have been addressed**: "I believe the ablation table in the rebuttal is very helpful, showing the contribution of SVD and activation separately". Unfortunately, the other reviewers were unable to continue the discussion due to current circumstances. We hope the AC will consider this positive engagement and resolution.


We briefly summarize the main concerns raised by the reviewers and how our rebuttal addressed them. Note that most concerns centered on the scaling laws and additional ablation studies, and only **Reviewer xG1E, rating 2** concerns about the novelty. Fortunately,  **Reviewer xG1E** engaged with us and indicated most concerns have been addressed:

1. **Novelty** (**Reviewer xG1E**). In our rebuttal, we clarified the novelty of LOST. First, LOST codesigns a distinct low-rank plus sparse pre-training architecture, specifically designing a channel-wise sparse structure to support hardware acceleration. Second, we employ a complementary SVD-based initialization between the low-rank and sparse components, which significantly enhances pre-training stability and performance. Third, we incorporate an activation function within the MLP to further enhance model representation capacity. These innovations allow LOST to outperform strong baselines and even match full-rank models. We have demonstrated this novelty through detailed analysis and empirical evidence.

2. **Scaling Laws** (**Reviewers a9CA & xG1E**). Reviewers raised concerns about whether LOST maintains its performance advantage during training with more data (e.g., 2x Chinchilla Law) and whether extended training makes initialization less important. To address this, we added experiments training a 1B model on 2x Chinchilla Law data and a 60M model on up to 7x Chinchilla Law data. The results show that LOST continues to perform strongly even with extended training data. Crucially, our findings confirm that initialization remains fundamental. The impact of initialization cannot be eliminated simply by extending training data (which would also inevitably increase training costs). These findings directly address the reviewers’ concerns.

3. **Additional ablations** (all reviewers). Reviewers requested more empirical results. During the rebuttal period, we added **all requested experiments**, including:
- Retraining the 7B model with more hyperparameter setting, which addresses concerns about fairness and clarifies that LOST maintains strong performance at scale.
- Additional experiments on FineWeb-Edu datasets, which address questions about dataset generalization and demonstrate that LOST performs well beyond C4.
- Pre-training results on more model architectures, i.e., GPT-2 and T5. This validates the generalizability of LOST across different model architectures beyond LLaMA.
- Comparisons of low-rank models using Kaiming and SVD initialization. This isolates the specific performance gains attributed to our co-design rather than SVD initialization alone.
- GLUE fine-tuning results for LOST, which address questions about downstream transfer and confirm that LOST maintains strong generalizability.
- Ablations on sparse-component initialization, which address why we select channels from the original matrix and verify that this choice leads to measurable improvements.
- 1B experiments with 2× Chinchilla data. This shows that LOST remains effective when trained with significantly more data.
- 60M experiments with 7× Chinchilla data, which further address the role of initialization and demonstrate that strong initialization remains crucial even with extremely long training duration.

These results fully validate the innovation and superior performance of LOST. We believe our rebuttal **fully addressed** all reviewer concerns.

While we regret that we are not able to hear the reviewers' final feedback on our detailed responses, we hope that these clarifications, the demonstrated effectiveness of our method, and particularly the constructive engagement from Reviewer xG1E, who acknowledged that our ablations successfully addressed their initial concerns, warrant a positive reassessment of our work.

Best, Authors

---

### Meta-Review · Area_Chair_yYhi · 2026-01-03

**Summary:**

This paper proposes LOST, a pre-training framework that combines low-rank parameterization with structured channel-wise sparsity through an SVD-based initialization. Before the rebuttal, reviewers have concerns regarding novelty, scaling laws, and design choices of different components. In the rebuttal, the authors made efforts to add additional experiments and justifications regarding these concerns.

After carefully reading the rebuttal and the paper, the Area Chair believes some concerns are not fully resolved. Specifically, the added ablation studies are conducted mostly at small scales (60M or 130M models). These experiments (component analysis on the LoRA Baseline and the study of $\gamma$.) should be scaled up to larger models to more convincingly demonstrate their effectiveness.

In addition, a central concern from reviewers is why the sparse and low-rank model outperforms the dense model. Although the authors provide several explanations, these justifications are not fully convincing, especially given the additional results on the FineWeb-Edu dataset. Unlike the results reported in the main paper, where LOST outperforms the dense model, LOST consistently underperforms the dense baseline on FineWeb-Edu. These results suggest that LOST may primarily function as a form of regularization or noise mitigation, and that its apparent performance advantages may come from the lower dataset quality of C4 relative to FineWeb-Edu, rather than from a clear advantage of the proposed method itself. The Area Chair does not intend to penalize the submission for providing additional results; rather, they believe this is a valuable problem and addressing it would largely increase the quality and potential impact of the paper.

Considering all the aforementioned factors, the Area Chair recommends rejection of this submission and encourages the authors to incorporate the reviewers’ suggestions to further strengthen the paper, particularly by exploring training on higher-quality datasets.

**Reviewer Concerns:**

The initial reviewer's concern includes novelty, scaling laws, and design choices of different components. After the rebuttal, the remaining concerns include design choices on larger-scale models and the impact of the training dataset quality.

**Reviewer Scores:**

Reviewer xG1E may increase their score to 4. Other reviewers would keep their original scores due to remaining concerns.

---

### Decision · Program_Chairs · 2026-01-26

Reject